# Preferences for accessing sexual and reproductive health services among adolescents and young adults living with HIV/ AIDs in Western Kenya: *A qualitative study*

**Harriet Fridah Adhiambo**[1☯]*, **Musa Ngayo**[2], **Zachary Kwena**[1☯]

**1** Centre for Microbiology Research, Kenya Medical Research Institute, Kisumu, Kenya, **2** Centre for Microbiology Research, Kenya Medical Research Institute, Nairobi, Kenya

☯ These authors contributed equally to this work.

\* fridahadhiambo@gmail.com

**Data Availability Statement:** All relevant data are within the paper and its Supporting Information files and the URL below https://figshare.com/account/home#/data.

## Abstract

Despite the need, adolescents and young adults (AYAs) in resource-limited settings have limited access to sexual and reproductive health (SRH) care services for improved health outcomes. This is worse for AYAs living with HIV in resource-limited settings where much is unknown about contexts and issues inhibiting access to SRHs. We explored adolescents', healthcare workers, and caregivers' preferences for access to sexual and reproductive health services for adolescents and young adults living with HIV. We conducted 30 in-depth interviews and 8 focus group discussions among a subset of AYA aged 14–24 living with HIV, healthcare workers, and caregivers/parents. We recruited participants from Lumumba Sub-County Hospital (KLM) and Kisumu County Referral Hospitals in Kisumu County (KCH). Trained and experienced qualitative research assistants 5–10 years older than the adolescents conducted interviews and facilitated discussions using guides designed to elicit detailed views and perspectives on sex and sexuality, access to SRH services, challenges of AYA living with HIV, and potential interventions to improve access to SRH services. Audio files were transcribed verbatim and translated to English where necessary before coding and analysis. We applied constant comparative analysis for theme and content to arrive at our conclusions. Our analysis yielded two main themes: preferences for a venue for SRH services and choices for qualities of an SRH counsellor. We found that AYAs generally preferred receiving SRH services to be co-situated within clinical facilities. We also observed gender differences in the qualities of SRH providers, with male AYAs preferring older male service providers compared to females who preferred younger female providers close to their age. The study highlighted the preferences of AYAs for accessing SRH, which need to be considered when designing their health programs. Further, AYAs seem to endite health systems to individualize access to SRH for AYAs living with HIV by providing a combination of attributes that meet individual preferences.

**Funding:** This study was an internally funded at Kenya Medical Research Institute. MON is the author who received the award. Sponsors and funders did not play a role in the study design, data collection and analysis, decision to publish, or preparation for the manuscript. 1. HAF was a Research Assistant in the project. Her roles included conducting participant interviews, transcription, coding and analysis. 2. ZK and MON- Principal Investigators ZK and MON provided oversight of research development and implementation. ZK was directly involved in leading and supervising the qualitative activities including focus group and individual interview guide development, conducting interviews, coding/ analysis and results dissemination. MON was involved in the management of local regulatory supervision and administration of the study. All authors reviewed and approved the manuscript.

**Competing interests:** The authors have declared that no competing interests exist.

# Introduction

Adolescents and young adults (AYA) constitute one of the highest risk groups for HIV infection and pregnancy. Globally, about a third of all new HIV infections occur among AYA aged 15–24 years, and over 20 million girls aged (15–19 years) get pregnant annually [1–3]. Sexual and reproductive health complications increase morbidity and mortality rates, making them the leading cause of death among AYAs 15–19 years. Despite considerable efforts in ensuring universal access to sexual and reproductive health care services aimed at attaining gender equality, AYA's access to these services and information about them is still a global challenge [4–6].

The World Health Organization (WHO) recommends ensuring adolescents have universal access to sexual and reproductive health services and rights: comprehensive sexuality education, contraception, safe abortion, condoms, cervical cancer screening and treatment, sexually transmitted infections (STI) treatment, and antenatal and postnatal services [7, 8]. However, access is still unsatisfactory, particularly in sub-Saharan Africa, thereby increasing the risk of poor health outcomes [9–11]. Studies in sub-Saharan Africa have consistently demonstrated a significant burden of unsafe abortions and unmet family planning needs among AYAs attributable to inadequate access to SRH services [12–14]. These SRH challenges are worse among sexually active AYAs living with HIV. They engage in high-risk sexual behavior, yet the existing policies, programs, and services are inadequate in responding to their sexual and reproductive health needs [15].

Several factors have been associated with inadequate access to sexual and reproductive health among AYAs. Some relate to a lack of knowledge, socio-cultural norms, religious beliefs, and cost and distance to hospitals [4, 9, 12, 16–18]. Other factors cited are health system-related such as clinic operating hours, lack of integration of services, and healthcare provider attitudes [4, 19–21]. For instance, a study conducted in Abawale District, Northwest Ethiopia, revealed that less than half of AYAs (41.2%) accessed and utilized SRH services at the hospitals because of these unfavourable factors [13]. Although studies have revealed that SHR needs among AYA infected with HIV and non-infected AYA are similar, AYA with HIV face even more significant challenges with access to SRH services [22]. They experience stigma, difficulties with disclosure to their sexual partners and negotiations for safer sex [15, 23]. Additionally, it has been noted that some providers are not comfortable discussing sexual matters with adolescents living with HIV/AIDs despite these adolescents being in HIV care and treatment for a while [24]. MCarrahera et al., while examining the reproductive health needs and experiences of adolescents living with HIV in Zambia, found out that healthcare providers were hesitant to talk to adolescents about sex and contraception due to cultural norms that uphold sex in marriage [25]. These factors need to be addressed for many sub-Saharan African countries to achieve sustainable development goals.

Like many countries in SSA, Kenya still grapples with challenges in improving the utilization of adolescent sexual reproductive health services that are urgently needed [13, 26]. To adequately address these challenges, AYA and other stakeholders' perspectives need to be understood and taken care of in the design of targeted and responsive interventions [27]. While most studies in this area have focused on adolescent knowledge, needs, barriers, and interventions regarding SRH among adolescents there is a dearth of data on contextual issues inhibiting access to SRHs among AYAs living with HIV [12, 18]. Documenting and triangulating the preferences of AYAs themselves with those of their healthcare providers and caregivers provides a formidable dossier to guide the design of responsive SRH interventions. Thus, we qualitatively explored the preferences of adolescents, healthcare providers, and caregivers regarding access to sexual and reproductive health services for adolescents and young adults living with HIV in Kisumu, western Kenya.

## Materials and methods

### Study design

This sub-study was a qualitative part of a mixed-methods study to assess multi-disciplinary approaches to individualize antiretroviral therapy to enhance adherence and improve health outcomes among adolescents and young adults in resource-limited settings. The study occurred between May 14 and September 5, 2019, within two hospitals in Kisumu County, supported by Family AIDS Care and Education Services (FACES).

### Study population, setting, and sample size

Our population consisted of adolescents, and young adults enrolled in HIV care at Lumumba Sub-County and Kisumu County hospitals and their healthcare providers and caregivers. The two are public hospitals with HIV care programs supported by FACES and have a combined HIV patient population of 14,661, with 9.5% being AYAs. FACES is a PEPFAR-funded HIV treatment and care program co-led by the Kenya Medical Research Institute and the University of California San Francisco. PEPFAR operates within hospitals owned by the Ministry of Health and operated by partners supported by PEPFAR funding.

FACES program supports health issues at the clinics and in the community using a family model of care. The health challenges tackled include HIV, TB, pediatric and adolescent health and growth, reproductive health, poverty, stigma reduction, and overall well-being. The program currently supports 61 clinics spread across Kisumu County. We conducted 30 IDIs and 8 FGDs in the participant's preferred language (either English, Swahili, or Dholuo) to reach the theoretical saturation of our main themes.

### Sampling procedures

We conducted in-depth interviews and focus group discussions among a subset of adolescents and young adults (AYA) aged 14–24 years, healthcare workers, and caregivers/parents. Adolescents and young adults were eligible if they were aged 14–24, living with HIV and on care at the two hospitals. Healthcare providers were eligible for FGDs if they were at least 18 years old and provided services to the AYAs at the patient support centres. Parents/caregivers were those identified by the AYAs and confirmed by the HIV care program staff. Research assistants purposively selected study participants to achieve variation in age, gender, and duration in care. This ensured that the study participants were knowledgeable and experienced in different aspects of SRH and HIV services. Research Assistants provided a list of potential participants (AYA and caregivers) generated from the reception module (An electronic database containing a list of all AYA receiving care and treatment at each clinic and contacts of caregivers) to the healthcare staff who approached the selected AYAs and caregivers (approached at the clinic during visits and over the phone), and informed them about the study, and asked if they were interested in participating. Healthcare workers were approached by the clinic in charge and informed about the study. Those interested were referred to our Research Assistants, who provided additional information and full disclosure about the study and consented to those willing to participate. Among 33 AYAs approached to participate in in-depth interviews, only 30 participated. Two were not interested, whereas one participant did not show up, and the reason was not determined. However, all participants mobilized for FGDs participated. Consenting participants were invited to participate in in-depth interviews or focus group discussions on a specified day and time. The study was mainly conducted during school breaks and some during their clinic appointment dates. Scheduling with other participants was based on their availability. Hence, participation in the interviews did not interfere with the participant's schedule/activities.

We obtained ethical approval to conduct this study from the Kenya Medical Research Institute's Scientific and Ethics Review Unit (KEMRI/SERU/CMR/00064/3528). We obtained written informed consent from all participants. Assent was obtained from participants below 18 years old, and their parents/caregivers provided written consent. Consenting and interviews were done in a private secluded room away from patient and clinical staff traffic at the adolescent HIV clinic. Consenting was done in the participant's preferred language (English, Swahili, or Dholuo). Participants were assured of their privacy and informed that discussions would be paused if there was any intrusion. Additionally, they were informed that their participation didn't affect the care they received at the HIV clinics. The study kept consent documents devoid of identifying information in a lockable cabinet within the study premises. We implemented our study following the ethical standards in the 1964 Declaration of Helsinki and its later amendments and other comparable ethical standards.

## Data collection

Research Assistants trained in qualitative methods with over three years of experience working with AYA in HIV clinics and SRH clinics conducted interviews and facilitated the discussions. Overall, the Research Assistants were 5–10 years older than the adolescents. We started the interviews and group discussions by collecting socio-demographic data and Research Assistants establishing rapport with participants. This ensured that participants were free to discuss sensitive issues that included conversations around sex and sexuality, for instance, the timing of sex and the use of condoms. We used in-depth interviews and FGD guides which covered topics on young people's sex and sexuality, access to sexual and reproductive health services (SRH), challenges AYAs living with HIV/AIDs face, and the potential interventions to improve access to SRH services. We had separate questions for the IDI and FGD guides for AYA and caregiver and health worker guides. Because of disclosure issues, FGDs were limited to general and group normative behavior, unlike IDIs that went deep into individual experiences. We conducted two FGDs with male adolescents, two with female adolescents, and one FGD held with boys and girls to get the discussion dynamics while together. The FGDs with health workers and caregivers combined both male and female participants. Both the IDI and FGD guides were available in English, Kiswahili, and the local Dholuo languages. The participants had a choice of which language they wanted the interview conducted. All interviews were audio-recorded with permission from participants, and interviewers took notes to back up recorded data. The notes also helped to remind interviewers to ask probes and follow-up questions. Audio files from the interviews were downloaded onto a password-protected computer, backed up on external drives, and kept off-site. The IDIs lasted approximately 40 while the FGDs went up to 90 minutes.

## Data management and analysis

Work on transcription and translation of the audio files started soon after the interviews. Audio files were transcribed verbatim and, where necessary, translated into English as the rest of the interviews continued. Once the transcripts became available, we started our thematic analysis by scanning through the transcripts to develop a codebook based on emerging broad and, eventually, fine codes. We used a combination of both inductive and deductive methods. Two people, the lead and the senior author, first collaboratively developed a codebook, identifying broad (themes) and fine (subthemes) codes. They independently coded the first four transcripts, compared code applications, discussed, and agreed on discrepant code applications, and refined the coding framework. Before analysis, the remaining transcripts were then divided up and coded by individual coders using the NVivo 12 qualitative data analysis

software (QSR International Pty Ltd, Melbourne, Australia). We used the constant comparative method to discover dominant themes that helped us understand the perspectives of adolescents', healthcare workers', and caregivers' perspectives on AYAs living with HIV access to sexual and reproductive health services and suggested interventions to improve access.

## Results

### Socio-demographic and behavioral characteristics

Most of our participants were female (57%), aged between 15 and 19 (73%). One-third had completed secondary education, while others had some secondary or college-level education. Regarding occupation, the majority (63%) were still in school, and one-fifth were unemployed. Only 16% reported employment, with 3% being self-employed. More than half of our participants (57%) had one/both parents deceased, with 27% with both parents deceased. Most participants (97%) were single, with 80% sexually active and only 7% having a pregnancy history. Most caregivers (92%) and healthcare workers were female, almost all above 24 years of age. Nearly half of the caregivers had college-level education, and 52% reported being self-employed (**Table 1**). The healthcare workers included were nurses (18.75%), clinical officers (18.75%), Pharmacy technologists (12.5%), Laboratory technicians (12.5%), Community Health Assistants (18.75%), and Peer Educators (18.75%). We did not observe any noticeable differences between the views of adolescent boys and girls, caregivers, and healthcare providers-subgroup analysis.

### Preferences of venue for receiving SRH services

The AYAs had differing perspectives on the ideal location for receiving SRH services. The venue to access the services varied from schools, health families, workplaces, churches, homes, youth-friendly centers, community halls, and hotels. Some of the critical considerations in the choice of venue to access SRH services were proximity, cost, waiting time at the venue, and concerns about privacy and confidentiality, as illustrated below:

> *At the facility. There is privacy and confidentiality here. Someone may not know what you have come to do (IDI KCH 004, 20-May-2019)*

> *At the hospital because the staff are very knowledgeable (IDI KCH 006, 20-May-19)*

> *I think anywhere would be good for a facility, support group, or workshop. Information is good from anywhere you get it; it could be a school where they could start a workshop as they are the most affected in terms of sexual reproductive health. So anywhere you provide the information will help (IDI KLM 033 5-Sep-19).*

> *Pharmacy because there are those people that stay far from health facilities and cannot afford to go there; therefore, going to the pharmacy would be ideal (P 8, AYA FGD at KCH).*

Notably, the most commonly voiced preference by AYAs in FGDs and IDIs was that SRH services be co-situated within the healthcare clinics. This is because they provide a one-stop shop for different services they may require. Having all the services needed under one roof saved them time, and one could confide in the health care provider. These sentiments were voiced by both AYAs, caregivers, and healthcare workers, as illustrated below:

> *No time wasting. You just come for the service and the drugs, then leave (IDI KCH OO6 20-May-19).*

**Table 1. Socio-demographic characteristics.**

| Attribute | Adolescents Count (%) | Caregivers Count (%) | Healthcare workers Count (%) |
|---|---|---|---|
| **Gender** | | | |
| Female | 17(57) | 12(92) | 9(56) |
| Male | 13(43) | 1(8) | 7(44) |
| **Age** | | | |
| 15–19 | 22(73) | 0 | 0 |
| 20–24 | 8(27) | 0 | 1(6) |
| Above 24 | 0 | 13(100) | 15(94) |
| **Level of education** | | | |
| College | 8(27) | 6(46) | 14(87) |
| Completed Secondary | 10(33) | 2(15) | 2(13) |
| Some secondary | 8(27) | 4(31) | 0 |
| Completed Primary | 0 | 1(8) | 0 |
| Some primary | 4(13) | 0 | 0 |
| **Occupation** | | | |
| Employed | 4(13) | 5(38) | 16(100) |
| Self-employed | 1(3) | 7(52) | 0 |
| Unemployed | 6(20) | 1(8) | 0 |
| In school | 19(63) | 0 | 0 |
| **Marital status** | | | |
| Married | 1(3) | 11(85) | 8(50) |
| Single | 29(97) | 2(15) | 8(50) |
| **Orphanhood status** | | | |
| Has both parents/caregivers | 13(43) | - | - |
| One parent deceased | 9(30) | - | - |
| Both parents deceased | 8(27) | - | - |
| **Sexually active?** | | | |
| Yes | 24(80) | 13(100) | 16(100) |
| No | 6(20) | 0 | 0 |
| **Ever been pregnant/expectant?** | | | |
| Yes | 2(7) | - | - |
| No | 28(93) | - | - |

*Yeah. You know if you do not get them at the same point. You know you have to move from one place to another, which is tiresome. The advantage of getting them at the same place is that you benefit twice (IDI KCH 002 18-May-19)*

*They will be very happy because once someone comes from the doctor's room to room 4, people will definitely know that you are going for planning services. When all the services are provided in one room, there is privacy (FGD with Caregivers at KCH, 29-Aug-19)*

*I think it will be awesome because we all want to go to a supermarket where we get everything, the one-stop shop. I think it will save time, and stigma issues will be addressed. Health providers should be trained on all manner of screening and offering other services (FGD with Healthcare providers at KLM, 15-May-19).*

Additionally, AYAs disclosed that they needed venues where they could quickly go to access SRH services under the pretext of seeking treatment for other conditions, such as malaria

without their parents suspecting since all the services would be offered in the same clinical room. These have been illustrated below:

> *This is because if I tell my parent that I am feeling unwell and want to go to the hospital, they will not deny me or start doubting. Once I get there then am the one who knows which services that I would like to access. This is because if I go to a private place, then she will have so many questions (FGD with adolescents at KCH)*

The participants further argued that hospitals also have trained and qualified staff who can address any issues they might have and give encouragement and support.

> *If you access those other things, maybe there are not good places you can get from them. But here, they will tell you the good ones and the bad ones. Some people teach you that you should use a condom, but they will not teach you how to open it or put it on, but here you are taught many things. (IDI #2 at KCH, the participant meant that when one accesses SRH services in a hospital, he/she will be advised accordingly by the trained staff compared to other locations.)*

However, AYAs who preferred other venues argued that they could access SRH services privately without people suspecting and stigmatizing them. To them, some hospitals are stigmatized, and people seen there automatically get labelled as being sick with HIV. AYAs reported that venues away from hospitals created a conducive environment that allowed them to freely discuss their SRH issues.

> *In church. I want somewhere where the pastor will open up with everyone like he should be courageous enough to tell me to do this and this and not the other things. It will be better for your own health. (IDI KLM 0123 16-Jul-2019).*

> *If I were told to choose, my ideal place would be events. For events, people would come first of all to look nice, but most importantly, no one will know whether you are HIV positive or negative. Again where there are events, trust me, youths will turn up in numbers. (IDI KCH 0014 22-May-2019).*

Some participants preferred receiving SRH services at retail pharmacies because of their strategic proximity to homes and no experience of long queues when they go for services. AYAs further explained that the pharmacies provided an opportunity for quick access to commodities such as condoms or e-pills in cases of emergency or unplanned activities.

> *You can get the other services in the hospital, but you can buy a condom from the shop. For example, if you have a girl and you don't have condoms, would you leave her to go to the hospital to buy condoms or instead you buy from the shop (FGD with adolescents at KCH on Aug 3 2019)*

Other AYAs didn't care much about the venue; they were all interested in accessing the services. Generally, older adolescents and young adults (18–24 years old) seemed to prefer receiving their SRH services at the hospitals.

## Preferences of qualities of SRH counsellors

AYAs, healthcare providers, and caregivers had varied views on who was best placed to counsel them on SRH issues. Some preferred healthcare providers, such as nurses and community

health volunteers (CHVs), while others tended towards religious ministers and teachers at school. One participant reported that it is usually challenging for parents to talk to their children about SRH issues, as done by healthcare providers such as CHVs.

> *I feel it would be beneficial because I have seen community health volunteers who talk to children and open up. I think the issue of age makes them have confidence in you (FGD with caregivers on 23-Aug-2019)*

Others were inclined to have a trained professional go to schools on specific days to provide SRH services and offer psychosocial advice. Others argued that if teachers were assigned to provide the services, they would be judgmental and deny AYAs services because they were young.

> *In a hospital, you are sure of getting it, you know when it is taken to schools like the teachers to be providers, some teachers will not allow you to access them since they will say you are not eligible due to your age, but maybe you know the reason why you want it (IDI# 16 at KCH).*

Regarding age, AYAs preferred providers who were their agemates because they said that it was easier for a young person to open up to such a peer provider. They argued that the same-age peer provider would understand what they are going through and speak the same language as someone older. However, they also pointed out that such a young provider should be well-trained and equipped with the correct information to pass over whenever necessary.

> *The child may also wonder why such an older person has been assigned to her. Due to the age difference, they might feel that this person will not know what she is undergoing. Healthcare [the young] providers should be trained so that they have the right information to deliver to these adolescents (FGD on 23-Aug- 2019)*

AYAs also pointed out that older people tend to misunderstand young people or ignore them altogether, thus creating an element of fear. However, others admired the seasoned knowledge, experience, and parental love that some older providers might exhibit towards young people.

> *I believe these older people have more experience than us but not all of them. You will just look at that person and see whether you can share. (IDI # 014 at KLM)*

> *I will just be comfortable because you know they understand what people pass through, you know he or she is an elder, and they know what they have passed through at various stages (IDI #013 at KCH)*

While some participants were comfortable discussing their SRH issues with people known to them, others held contrary opinions that they would be comfortable with strangers whom they might never meet again after the encounter. They preferred a stranger because they assumed they might not meet again and feel embarrassed about what they disclosed.

> *I would prefer someone who is my mother's age and who is a stranger to me. This is because after sharing, I will not see her, nor will she see me (FGD with adolescents at KCH on 28-Aug-2019).*

A combination of age and gender also came up strongly as attributes of the SRH service providers the AYAs preferred. This was observed among the AYAs, caregivers, and healthcare

providers during the IDIs and FGDs. While male participants tended to select an older person of the same gender, females were comfortable with close friends and relatives of the same age and gender. The reason for preferring older people of the same gender was that they have experience by virtue of having gone through some of the stages of life.

> *I will not be comfortable but rather be comfortable with someone much older and of the same gender as me. . . I will feel comfortable with someone who is older and is of my gender because they have passed through the stages I am passing through (FGD with adolescents at KCH on Aug 28, 2019)*

> *Because she will be older than me, she knows everything, and she has lived a long, she will be giving me some knowledge to know all about sexual reproductive health (IDI #031 at KCH).*

> *We are of the same gender, and there is nothing I can hide because maybe she has gone or has seen more than me. That is why I can be open and tell her everything so that if she has other advice or encouragement, she can share it with me (IDI # 035 KLM).*

Some healthcare providers and caregivers felt that due to the heterogeneity of young people as a demographic group, access to SRH should be individualized by providing a combination of attributes that meet different preferences. For instance, we can make available both older and younger as well as male and female, which allows AYAs to choose their preferences, such as older male, younger male, older female, and younger female.

> *I think like the government has tried to bring the youths on board so that it becomes peer-to-peer, but at the same time, we are looking at it as if you get an older provider and you are sharing one on one, then you get more sense of it. She's like a representative of your mother. If I get a younger person, we can just talk, and maybe, I will lie to that person. It's both positive and negative. The government should bring both the old and the young. You can choose who to see you. If you get a female or a male one (FGD with Healthcare providers at Lumumba Hospital on May 23, 2019)*

> *There should be an arrangement where the provider sees children of the same gender as the providers. Or also, there should be both male and female providers in the room at the same time (FGD with caregivers May 17, 2019).*

## Discussion

This study explored the perspectives of adolescents, caregivers, and health care providers on improving access to sexual and reproductive health services of AYAs living within western Kenya. We found that AYAs generally preferred receiving SRH services co-situated with clinical facilities. We also established that there were gender differences in the qualities of SRH providers. Male AYAs preferred older male service providers, whereas females preferred younger female providers close to their age.

Designing interventions responsive to AYAs' aspirations that seek to maximize the chances that such interventions would look appealing and taken up is critical. Part of the trick in designing such interventions is engaging and incorporating the views of AYAs themselves and running by their healthcare providers and caregivers for concurrence. Involving end-users in intervention designs has been successfully used in various contexts [23, 25, 26, 28]. For instance, Shrier et al., while applying systems thinking and human-centred design in

developing intervention implementation strategies to improve SRH access for AYAs with depression, established that employing user-centred designs can be used to generate design ideas and create prototypes for innovative program implementation that AYAs may like and use [23]. These findings undoubtedly demonstrate the importance of involving end-users and other critical stakeholders in the search for essential attributes to consider in the intervention design, as was done in this study.

One of our main findings was the need to design SRH interventions that ensure all services the AYAs require are *"under one roof."* Various reasons, including prohibitive distance, costs, waiting times, and even concerns about privacy and confidentiality, make AYAs prefer a one-stop shop where they obtain all services. Some PEPFAR HIV care programs, such as FACES, have developed youth centers that allocate their own space and services, including libraries and games. This innovation ensures that the youth are happy and comfortable when HIV care and other services address the age-old challenge of youth retention in care.

One of the challenges cited widely in the literature as a barrier to AYA access to SRH is the venue for these services. Many AYAs have reported challenges accessing some of the venues, begging the question of where the AYAs would want to access their services. Though with differing convictions recorded, we found that AYAs mostly preferred receiving their SRH services at hospitals; challenges associated with such public spaces notwithstanding. Other studies in sub-Saharan Africa also show similar preferences for AYAs receiving SRH services from hospitals [29, 30]. The main reasons are their widespread geographical location and the presence of trained providers.

Although most of the AYAs in our study strongly preferred receiving their SRH services at hospitals, some preferred other outlets like retail pharmacies. They argued that these venues provided quick and friendly services devoid of disrespectful staff attitudes and stigma, compromising AYA service uptake and retention [31, 32]. For instance, a study conducted in Lao People's Democratic Republic examining perceived barriers to accessing SRH services among adolescents demonstrated that young people utilized pharmacies often to evade hospital complexities such as long queues, counselling for SRH services, and the likelihood of being identified [4]. These findings highlight the importance of having a safe space environment for providing SRH services within HIV clinics and interventions that will improve access to the existing SRH service provided to adolescents living with HIV/AIDs at the adolescent centers.

Service provider characteristics influence the uptake and utilization of healthcare services offered. Similar to our findings, provider age and gender preferences have emerged from other studies as key attributes in providing SRH services to AYAs [33, 34]. Male AYAs in our study preferred older male service providers due to their seasoned knowledge and experience over time, whereas the female AYAs had preferences for female peer providers. We hypothesize that girls prefer younger healthcare professionals because they are approachable. The female AYAs are free to share with them and have recent experiences transitioning from adolescence to adulthood. In most cultures from different ethnic communities in Kenya, males are mentored on social roles and community cultures by older men; therefore, there is a likelihood that the male AYA would consider having an older man for their SRH needs. Napit et al., while examining factors associated with the utilization of adolescent-friendly services in Bhaktapur district, Nepal, found that adolescents had a same-sex preference for service providers [31]. These and other studies seem to point to the need for health systems to individualize access to SRH among AYA by providing a combination of attributes that meet different preferences for the AYA. This will ensure that AYAs are enthusiastic about accessing health services and thus can be monitored and started on required interventions promptly.

Two factors potentially limited this study; firstly, recall bias. Some questions required the adolescents to remember specific details retrospectively, which was challenging for some and,

therefore, might have compromised the details of the data obtained. Secondly, the study was limited to two hospitals in Kisumu County and, therefore, can only be applied in the local contexts due to variations in Kenya's social, cultural, geographical, and epidemiological contexts. Despite these limitations, our study triangulated information from AYAs with their healthcare providers and caregivers to obtain a complete picture of their perspectives and preference for accessing SRH services. In conclusion, AYAs generally preferred receiving SRH services to be co-situated within clinical facilities. However, we did not find significant differences between the AYAs/caregivers/healthcare workers on where SRH services were best provided. We also observed gender differences in the qualities of SRH providers, with male AYAs preferring older male service providers compared to females who preferred younger female providers close to their age. The study highlighted the preferences of AYAs for accessing SRH, which need to be considered when designing their health programs. Further, AYAs seem to endite health systems to individualize access to SRH for AYAs living with HIV by providing a combination of attributes that meet individual preferences. This information on AYA's preferences for accessing SRH services is critical in designing health programs responsive to their aspirations of individualized care.

## Supporting information

**S1 Appendix.**
(DOCX)

**S1 File.**
(DOCX)

**S2 File.**
(DOCX)

## Acknowledgments

We acknowledge the administrative support we received from the Director General of Kenya Medical Research Institute (KEMRI), the Director Center for Microbiology Research (CMR), as well as the Co-Directors of the Research, Care and Training Program (RCTP). We also thank the leadership of the Family AIDS Care and Education Services (FACES) leadership and the staff at Kisumu County Hospital and Lumumba Sub-County Hospital for their logistical support and collaboration. Finally, we appreciate Irene Adhiambo Okumu's invaluable contribution to recruiting participants, conducting in-depth interviews, and facilitating focus group discussions.

## Author Contributions

**Conceptualization:** Harriet Fridah Adhiambo, Musa Ngayo, Zachary Kwena.

**Data curation:** Harriet Fridah Adhiambo, Musa Ngayo, Zachary Kwena.

**Formal analysis:** Harriet Fridah Adhiambo, Musa Ngayo, Zachary Kwena.

**Funding acquisition:** Musa Ngayo.

**Investigation:** Musa Ngayo, Zachary Kwena.

**Methodology:** Musa Ngayo, Zachary Kwena.

**Project administration:** Musa Ngayo, Zachary Kwena.

**Resources:** Musa Ngayo, Zachary Kwena.

**Supervision:** Zachary Kwena.

**Validation:** Harriet Fridah Adhiambo, Musa Ngayo, Zachary Kwena.

**Writing – original draft:** Harriet Fridah Adhiambo.

**Writing – review & editing:** Harriet Fridah Adhiambo, Musa Ngayo, Zachary Kwena.

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
