## [Decision Letter · Decision Letter 0]

2 Aug 2022

PONE-D-21-29137

Preferences for accessing sexual and reproductive health services among adolescents and young adults living with HIV/AIDs in Western Kenya: A qualitative study.

PLOS ONE

Dear Dr. Adhiambo,

Thank you for submitting your manuscript to PLOS ONE. After careful consideration, we feel that it has merit but does not fully meet PLOS ONE’s publication criteria as it currently stands. Therefore, we invite you to submit a revised version of the manuscript that addresses the points raised during the review process.

While the reviewers agree that this is an important topic, there are some issues with the paper in its current form that need to be addressed. Notably, the reviewers question issues of confidentiality in your supplemental documents. In addition to addressing all other reviewer comments, please also address how the confidentiality of participants is protected. Please provide information on ethics board approval as well. Ensure that there are NO confidentiality breaches in any of the information provided within the document or in the supplemental contents (see reviewer #1 comments for an example of potentially identifiable information) as this paper cannot be published with ethical violations.

We look forward to receiving your revised manuscript.

Kind regards,

Bettye A. Apenteng

Academic Editor

PLOS ONE

Journal Requirements:

2. We note that you included minors (age<18) in your study. Please provide additional details regarding minors consent. In the ethics statement in the Methods and online submission information, please ensure that you have specified whether you obtained consent from parents or guardians. If the need for consent was waived by the ethics committee, please include this information.

4. Thank you for stating the following financial disclosure: "This study was an internally funded at Kenya Medical Research Institute.

MON is the author who received the award. Sponsors and funders did not play a role in the study design, data collection and analysis, decision to publish, or preparation for the manuscript."

We note that one or more of the authors is affiliated with the funding organization, indicating the funder may have had some role in the design, data collection, analysis or preparation of your manuscript for publication; in other words, the funder played an indirect role through the participation of the co-authors. If the funding organization did not play a role in the study design, data collection and analysis, decision to publish, or preparation of the manuscript and only provided financial support in the form of authors' salaries and/or research materials, please do the following:

a. Review your statements relating to the author contributions, and ensure you have specifically and accurately indicated the role(s) that these authors had in your study. These amendments should be made in the online form.

b. Confirm in your cover letter that you agree with the following statement, and we will change the online submission form on your behalf: 

“The funder provided support in the form of salaries for authors [insert relevant initials], but did not have any additional role in the study design, data collection and analysis, decision to publish, or preparation of the manuscript. The specific roles of these authors are articulated in the ‘author contributions’ section.

Reviewers' comments:

Reviewer's Responses to Questions

**Comments to the Author**

1. Is the manuscript technically sound, and do the data support the conclusions?

Reviewer #1: Partly

Reviewer #2: Yes

2. Has the statistical analysis been performed appropriately and rigorously? 

Reviewer #1: N/A

Reviewer #2: Yes

3. Have the authors made all data underlying the findings in their manuscript fully available?

Reviewer #1: No

Reviewer #2: Yes

4. Is the manuscript presented in an intelligible fashion and written in standard English?

Reviewer #1: No

Reviewer #2: Yes

5. Review Comments to the Author

Reviewer #1: Apologies to focus on the negatives, there was clearly good and necessary research in this document, but my comments are below. My biggest concerns are CONFIDENTIALITY BREACHES OF PEOPLE LIVING WITH HIV IN THE SUPPLEMENT document.

Ethics:

Two major points of concern:

1. The information in the supplementary information is inappropriate. I only read a tiny bit of it and already had a young person’s town, school name, their age, school year, employment, sex, and their intentions to go to university. IT IS THEREFORE POTENTIALLY ENTIRELY IDENTIFIABLE to anyone who knows these young people. This error makes me question data governance in this project. In my opinion, local actions need to be taken to investigate this confidentiality breach. The interview transcripts should never in my opinion be published in their current form. A suitable alternative would be to append the interview and focus group discussion topic guides.

I think that the interview and focus group topic guides should be appended rather than the interviews themselves in the form that they are in.

2. Why were healthcare workers and caregivers asked if they were sexually active? This does not seem an appropriate question as the study was NOT about their sexual behaviour. Was the ethics team aware that this would be asked of these groups of people? In my country this would be inappropriate data to obtain in this context, as it does not relate to the research question. Again, I question whether training on data governance needs to be strengthened in the reporting team.

General feedback:

The use of the word "facility" throughout this piece had me confused and needs some clarity.

I would want the interview and FG topic guides as appendices (and not the transcripts, for reasons above)

Abstract:

“AYAs generally preferred receiving SRH services from the health facilities offered and served by trained healthcare providers.” – having read the whole paper, I am not sure I have seen evidence of the “trained healthcare providers” bit of this statement? I think maybe the abstract should just say that AYAs generally preferred SRH services to be co-situated with clinical facilities? Your data seems to support the idea that healthcare providers feel that training is important.

Introduction:

A Key issue: I think for me the article would be a bit easier to understand if a bit more of the context was given about the presence of PEPFAR and its projects and centres for AYA living with HIV, their relationship to state-provided healthcare, and therefore the policy need for research of this type.

“Like many countries in SSA, Kenya still grapples with challenges in improving the utilization of adolescent sexual reproductive health services that urgently need focus” – Is focus a big issue for healthcare providers? I think this needs to be made clearer

The other clear difference between AYAs living with and living without HIV is that the latter group is already in regular contact with healthcare professionals for their HIV. In many countries these professionals would feel confident in advising the AYA about SRH needs, certainly in giving out condoms, or signposting them to SRH-specialist services. The challenges here are that some services may have known the AYA since they were a child, and then the child feels that they cannot discuss their sexuality with them, and that the HIV team will have an interest in discussing disclosure with the AYA. A fresh or unknown staff member from an SRH team can be helpful in these situations. I didn’t hear much about this angle from the introduction and would like to know how things are in Kenya and why the patients’ own HIV teams aren’t able to cover their SRH needs. Or are they? Are they meant to? It would be really useful background information to have, as then the other questions and statements would make more sense.

Methods:

Unclear how patients were consented, especially regarding the language they were consented in, language of consent form, and were the participants clear that participation didn’t affect the care they were receiving, as PLWHIV? I note some people had only “some primary” education – were measures put in place to support those who were illiterate?

Data security of recordings etc is not mentioned?

Unclear in methods section why broad code groups were given in this section– would usually be in results section unless the codes were a priori codes, furthermore, it is not clear if these codes were a priori codes or not and this needs to be clarified.

Did the interviewer sex match the participant’s? what training had the research assistants had? Abstract mentions they were trained and experienced but this isn’t in the text, which is a bit unusual as typically there shouldn’t be anything in the abstract that isn’t inside the main article. Training is especially important as discussing sex with very young people.

Also, how old were the interviewers, given your research’s findings that young people prefer to talk to younger staff?

Were the FGDs held with boys and girls together – if so, it would be interesting to know what drove this decision?

Where were interviews held? What efforts were made to keep the conversations from being heard? Was anyone else present or allowed to be present when the IDIs were held? Were parents informed of the interviews? how were issues such as school attendance navigated for these interviews, to encourage participation and minimise impact on life?

Were there any pilots performed? If so, should be mentioned.

What proportion of those approached agreed to participate, if known? If the number of people invited is unknown, this should be stated, and why it is unknown.

I am not clear whether the AYAs and the carers were known to each other or not, ie. were the carers picked because they cared for the specific AYA being interviewed?

How were carers and healthcare workers approached to be in the study, I couldn’t see this?

How was the number 30 chosen? I am not clear at what point saturation was found – was it after all 30 had been interviewed or before then? The wording isn’t quite clear. It reads more like 30 was chosen as it was expected that by this point saturation would have been reached, which is not quite the same thing as reaching thematic saturation.

Were all 3 FGDs asking the same questions, or was each group tackling different questions? Also, how many people were in each FGD?

Results -

What groups were the healthcare workers from? Eg. what proportion was nursing, doctors, etc. what proportion of the patients were living with HIV? And what proportion of the patients participated in group discussions? Did all the AYAs have IDIs and FGDs?

At least one male in the staff group seems to have (perhaps understandably as presumably he was indicating that he is a father) said that he was previously pregnant/expectant – this raises questions about whether the question was relayed to all participants in the same way? It is probably worth unpicking from your data and commenting to explain how this happened (or changing the way the question is described in the table).

I would have liked a general paragraph in the results to outline how the FGDs had been – was there discord? How much did people disclose? Did boys or girls, older or younger people dominate discussions in the FGD for AYAs? Were staff in general agreement or were there differences eg. based on rank or location of their work? Was there a lot of difference in the views raised at the 3 groups? Were the caregivers able to give many opinions? What were the main concerns within the 3 groups of people?

Why are the IDIs so under represented in your quotes? It seems a shame that you did ?30 IDIs and they only get 2 quotes.

Preferences sections:

Were the locations discussed (schools, hotels, pharmacies etc) prompted from a list/interview or FG prompt or spontaneously mentioned by the participants in the different groups?

“If you access those other things, maybe there are not good places you can get from them. But here, they will tell you the good ones and the bad ones (IDI #2 at KCH)” I think this quote needs some extra words in square brackets to help it make sense as at the moment I don’t understand it, even within the context given in the text.

“I feel that it would be beneficial because I have seen community health volunteers who talk to children, and they open up. I think the issue of age makes them have confidence in you (FGD on 23 August 2019” – this quote needs to say which of the 3 FGD groups this quote came from

“Others were inclined to have a trained professional go to schools on specific days to provide SRH services and offer psychosocial advice. However, other participants had a contrary opinion arguing that if teachers were to be assigned to provide the services, they would be judgmental and deny AYAs services with the reasoning that they are young and they should not be accessing such services.” - I do not find these two statements contradictory unless one assumes that the trained professional going to the schools is a teacher (and I don’t know why they would necessarily be?). there is also quite a lot of duplication of this paragraph in the following quotation so it could be trimmed.

“elderly” doesn’t really feel like the correct word, I think, where you have used it. Perhaps just “older” would be better?

Throughout the section on preferences of counsellors I wasn’t getting a very good idea of which groups of people were thinking what in terms of the preferences. This needs a bit more clarity (even if they were all in agreement, for example).

“receiving services at the facility” – this could be clearer eg. by stating “within a hospital building” (as to me, facility could mean anything including a school for example). I think this terminology is in the abstract as well, and would just be clearer if it used more precise language.

Key point: Did the IDIs concur up with the FGDs on your key findings (ie. preferred gender and age of care provider for girls and boys, and preference for SRH services co-situated at a venue like a clinic or hospital)? Did the different FGDs generally concur with this or were some more in agreement with your conclusion and others less so? I think this needs a bit more fleshing out as it is interesting if there are trends (rather than just the odd quote).

Discussion:

I don’t feel from reading your results that there was a strong prefererence for hospital or clinic co-situated SRH facilities, among the AYAs; rather that there was a large mix of views and maybe the predominant view was one of reference for co-situated SRH services (the word you use is “many”). I feel the discussion and abstract implies there was a fairly clear preference for healthcare co-situated SRH services. Therefore I suggest that either the results section needs to justify the conclusion that there was an overall preference (eg. by stating that a majority of participants said it (please also see above note about showing more clearly if it was in FGD or in IDIs that this conclusion was most evident; AYAs or staff/caregivers)), or the discussion/abstract need to decrease the weight currently placed on preference for SRH being co-situated with healthcare (eg by saying “the most commonly voiced preference by AYAs in ___(FGD/IDIs) was x”).

“AYAs generally preferred receiving SRH services from the health facilities offered and served by trained healthcare providers.” – having read the whole paper, I am not sure I have seen evidence of the “trained healthcare providers” bit of this statement? I think maybe the abstract should just say that AYAs generally preferred SRH services to be co-situated with clinical facilities? Your data seems to support the idea that healthcare providers feel that training is important.

“One of our main findings was the need to design SRH interventions that ensure all services the AYAs require are “under one roof."” . This may be true but you don’t evidence it much if it is a main finding. For example, there are no quotes from AYAs illustrating this (the quote that mentions parents not knowing what services are being sought does not illustrate the desire for a “one stop shop” with multiple services for the AYA, merely the fact that a health facility masks the SRH activity by also offering non-SRH activity that the person does not intend to use).

The sentences about the youth centres are interesting, but I don’t really see how they link in with the “under one roof” idea very clearly – are they co-sited with hiv clinical teams? This needs greater clarity.

I’m not sure what a “stigmatizing outburst” is – needs greater clarity/rephrasing.

“These findings highlight the importance of having a ‘safe space’ environment for the provision of SRH services within our health facilities and interventions that will improve access to the existing SRH service provided to adolescents living with HIV/AIDs at the adolescent centers.” Who does “our” refer to? And where would the safe spaces be? Within PEPFAR or within the health centres?

It would be interesting to hear why you think that girls preferred younger, and men older, healthcare professionals – is there any theory you can link with?

Typos:

AYA, SRH, SSA, IDI, FGD – define when first mentioned, not later in the document

“20 million girls aged (15-19 years)” should be “girls (aged 15-19)”

“if they were aged 14-24, living with HIV and on care at the two facilities” [should read IN care]

I can see that orphan/total orphan is a recognised term but it is a bit confusing – suggest it might help readability to change to “one/both parents deceased”

“This ensured that participants were free to discuss sensitive issues” – suggest you might mean “felt comfortable”?

Not sure if a typo – what are “health families” p.15 on my document?

“This study explored the perspectives of adolescents, caregivers, and health care providers on improved access to sexual and reproductive health services of AYAs living within western Kenya.” Currently this reads as though the services are already improved and people are being asked about them – I’m not sure this is what you mean though.

“horrible staff attitude” – suggest you mean “disrespectful staff attitudes” or something similar. “Horrible” is not an appropriate word for a journal, unless a quotation.

“In as much as AYAs in our study had a strong preference for receiving their SRH services at health facilities, some preferred other outlets like retail pharmacies” – this doesn’t make sense to me linguistically and I think you mean

“Although most of the AYAs in our study had a strong preference for receiving their SRH services at health facilities, some preferred other outlets like retail pharmacies”.

Reviewer #2: Well written and excellent manuscript of an important issue around AYA. The introduction however needs to be revised on various aspects- grammar, references and literature. The factors outlined as contributing to inadequate access to SRH do not only apply to ALHIV but to all adolescents in general. The distinction between these two groups and the differences need to be outlines as well as the possible reasons. More literature is needed in the background regarding preferences of healthcare providers and caregiver. More results could be included as well. Also in the discussion comparison with other studies could be made stronger.

6. PLOS authors have the option to publish the peer review history of their article (what does this mean?). If published, this will include your full peer review and any attached files.

Reviewer #1: No

Reviewer #2: No

---

## [Author Response · Author response to Decision Letter 0]

10 Sep 2022

https://journals.plos.org/plosone/s/file?id=wjVg/PLOSOne_formatting_sample_main_body.pdf
https://journals.plos.org/plosone/s/file?id=ba62/PLOSOne_formatting_sample_title_authors_affiliations.pdf

We have revised the manuscript to meet PLOS ONE’s style requirements including file naming.

2. We note that you included minors (age<18) in your study. Please provide additional details regarding minor’s consent. In the ethics statement in the Methods and online submission information, please ensure that you have specified whether you obtained consent from parents or guardians. If the need for consent was waived by the ethics committee, please include this information.

We have provided additional detail regarding minor’s consent. Assent was obtained for adolescents below 18 years, and their parents/caregivers provided written consent (Line 189-190). 

We have updated our submission based on the PLOS LaTeX template.

4. Thank you for stating the following financial disclosure: "This study was an internally funded at Kenya Medical Research Institute.

MON is the author who received the award. Sponsors and funders did not play a role in the study design, data collection and analysis, decision to publish, or preparation for the manuscript."

We note that one or more of the authors is affiliated with the funding organization, indicating the funder may have had some role in the design, data collection, analysis or preparation of your manuscript for publication; in other words, the funder played an indirect role through the participation of the co-authors. If the funding organization did not play a role in the study design, data collection and analysis, decision to publish, or preparation of the manuscript and only provided financial support in the form of authors' salaries and/or research materials, please do the following:

a. Review your statements relating to the author contributions and ensure you have specifically and accurately indicated the role(s) that these authors had in your study. These amendments should be made in the online form.

Below are the author/co-author roles. These changes have been updated in the online submission form.

1. HAF was a Research Assistant in the project. Her roles included conducting participant interviews, transcription, coding and analysis.

2. ZK and MON-Principal Investigators

ZK and MON provided oversight of research development and implementation. ZK was directly involved in leading and supervising the qualitative activities including focus group and individual interview guide development, conducting interviews, coding/analysis and results dissemination. 

MON was involved in the management of local regulatory supervision and administration of the study. 

All authors reviewed and approved the manuscript.

b. Confirm in your cover letter that you agree with the following statement, and we will change the online submission form on your behalf: 

“The funder provided support in the form of salaries for authors [insert relevant initials] but did not have any additional role in the study design, data collection and analysis, decision to publish, or preparation of the manuscript. The specific roles of these authors are articulated in the ‘author contributions’ section.

We agree with the statement, and we have updated the cover letter to reflect this.

We initially uploaded transcripts and would want to request if they could be withdrawn so that we upload the in-depth and focus group discussion guides as suggested by Reviewer #1. 

Thank you

We have uploaded the files containing in-depth interviews and focus group discussion transcripts. Should our manuscript be accepted for publication, we allow the journal to use these files as repositories.

7. Please include captions for your Supporting Information files at the end of your manuscript, and update any in-text citations to match accordingly. Please see our Supporting Information guidelines for more information: http://journals.plos.org/plosone/s/supporting-information

Reviewers’ Comments

Reviewer #1: Apologies to focus on the negatives, there was clearly good and necessary research in this document, but my comments are below. My biggest concerns are CONFIDENTIALITY BREACHES OF PEOPLE LIVING WITH HIV IN THE SUPPLEMENT document.

Ethics:

Two major points of concern:

1. The information in the supplementary information is inappropriate. I only read a tiny bit of it and already had a young person’s town, school name, their age, school year, employment, sex, and their intentions to go to university. IT IS THEREFORE POTENTIALLY ENTIRELY IDENTIFIABLE to anyone who knows these young people. This error makes me question data governance in this project. In my opinion, local actions need to be taken to investigate this confidentiality breach. The interview transcripts should never in my opinion be published in their current form. A suitable alternative would be to append the interview and focus group discussion topic guides.

I think that the interview and focus group topic guides should be appended rather than the interviews themselves in the form that they are in.

We apologize. This was an oversight on our end. 

We mistakenly uploaded our raw files instead of separate transcripts that did not contain participant identifying information. Based on the reviewer’s advice we are withdrawing the transcripts and instead submitting the interview guides.

2. Why were healthcare workers and caregivers asked if they were sexually active? This does not seem an appropriate question as the study was NOT about their sexual behavior. Was the ethics team aware that this would be asked of these groups of people? In my country this would be inappropriate data to obtain in this context, as it does not relate to the research question. Again, I question whether training on data governance needs to be strengthened in the reporting team.

We apologize if this was not clear. The question on sexually active was meant for adolescents and not healthcare workers

General feedback:

The use of the word "facility" throughout this piece had me confused and needs some clarity.I would want the interview and FG topic guides as appendices (and not the transcripts, for reasons above)

 In Kenya, hospitals are commonly referred to as health “facility.” We have replaced the word “facility” with HIV clinic or hospital based on the context.

Abstract:

“AYAs generally preferred receiving SRH services from the health facilities offered and served by trained healthcare providers.” – having read the whole paper, I am not sure I have seen evidence of the “trained healthcare providers” bit of this statement? I think maybe the abstract should just say that AYAs generally preferred SRH services to be co-situated with clinical facilities? Your data seems to support the idea that healthcare providers feel that training is important.

Thank you for this comment. We have revised the abstract and the discussion to reflect preference of SRH services to be co-situated with clinical facilities (line 19).

Introduction:

A Key issue: I think for me the article would be a bit easier to understand if a bit more of the context was given about the presence of PEPFAR and its projects and centers for AYA living with HIV, their relationship to state-provided healthcare, and therefore the policy need for research of this type. 

We gave some context on PEPFAR in the study setting section since the hospitals in which we conducted the study are supported by PEPFAR-funded HIV care programs. 

We have revised the section to include additional roles and the relationship with state-provided healthcare (line 139-140)

“Like many countries in SSA, Kenya still grapples with challenges in improving the utilization of adolescent sexual reproductive health services that urgently need focus” – Is focus a big issue for healthcare providers? I think this needs to be made clearer.

We have clarified in the introduction section that the SRH services need urgent focus by stakeholders engaged in adolescent sexual and reproductive health (Line 87-89).

The other clear difference between AYAs living with and living without HIV is that the latter group is already in regular contact with healthcare professionals for their HIV. In many countries these professionals would feel confident in advising the AYA about SRH needs, certainly in giving out condoms, or signposting them to SRH-specialist services. The challenges here are that some services may have known the AYA since they were a child, and then the child feels that they cannot discuss their sexuality with them, and that the HIV team will have an interest in discussing disclosure with the AYA. A fresh or unknown staff member from an SRH team can be helpful in these situations. I didn’t hear much about this angle from the introduction and would like to know how things are in Kenya and why the patients’ own HIV teams aren’t able to cover their SRH needs. Or are they? Are they meant to? It would be really useful background information to have, as then the other questions and statements would make more sense.

From a review of the literature, SRH staff members are hesitant to discuss with AYA living with HIV/AIDs matters on sex and contraceptives due to cultural norms. The same applies to some healthcare workers in Kenya, they believe AYA are not supposed to engage in sex and or use contraception before they are married hence AYA feel they are judged with these healthcare providers when they access SRH services. Indeed an unknown staff member would be helpful in breaking this barrier particularly among AYA who have been in care from. We have revised paragraph 3 of the introduction to reflect some of these ideas (Line 80-85).

Methods:

Unclear how patients were consented, especially regarding the language they were consented in, language of consent form, and were the participants clear that participation didn’t affect the care they were receiving, as PLWHIV? I note some people had only “some primary” education – were measures put in place to support those who were illiterate?

Participants were consented in their preferred language i.e. either English, Swahili, or Dholuo and assured that their participation in the study would not interfere with the care they were receiving at the HIV clinics. We have updated line (192-195) in the manuscript. 

Despite just having some primary education, participants were able to comprehend the questions being asked in the language of their choice and responded well to the questions. The research assistants clarified questions in cases where participants seemed not to have a clear understanding of what was asked. We did not have cases of illiterate participants’ who needed extra support. 

Data security of recordings etc. is not mentioned.

We indicated in the manuscript (Line 232-234) that the audio files from the interviews were downloaded onto a password-protected study computer and the same backed up on external drives and kept off-site with restricted access. 

Unclear in methods section why broad code groups were given in this section– would usually be in results section unless the codes were a priori codes, furthermore, it is not clear if these codes were a priori codes or not and this needs to be clarified.

We used a combination of inductive and deductive methods to derive the codes and the codebook which means some of the codes were a priori. We have updated this information in line 241-242 in the manuscript and omitted the broad code groups earlier mentioned. 

Did the interviewer sex match the participants? what training had the research assistants had? Abstract mentions they were trained and experienced, but this isn’t in the text, which is a bit unusual as typically there shouldn’t be anything in the abstract that isn’t inside the main article. Training is especially important as discussing sex with very young people.

Thank you for this comment. The research assistants were trained in qualitative methods and handling adolescents and had accumulated more than 3 years’ experience working with adolescents and young adults at the SRH clinics. We have revised line 202-204 in the manuscript to reflect this. 

 Also, how old were the interviewers, given your research’s findings that young people prefer to talk to younger staff? 

The interviewers were 5-10 years older compared to the participants but were well trained and experienced in handling adolescents including, establishing rapport and making adolescents comfortable to talk. 

Were the FGDs held with boys and girls together – if so, it would be interesting to know what drove this decision? 

We conducted two FGDs with male adolescents, two with female adolescents, and one FGD held with boys and girls together to get the dynamics of the discussion while together. The FGDs with health workers and caregivers had a combination of both male and female participants. Our experienced moderators managed the discussions to minimize dominance within individual participants as well as between gender (i.e. males or females) 

Where were interviews held? What efforts were made to keep the conversations from being heard?

The interviews were held in private and secluded rooms away from patient and clinical staff traffic. The participants were assured of their privacy and informed that they will pause discussions if there was to be any intrusion (191-195).

Were parents informed of the interviews? how were issues such as school attendance navigated for these interviews, to encourage participation and minimize impact on life? 

Parents/ caregivers of participants below 18 years provided consent for their children’s participation in the study and were involved in scheduling of interviews of their children. The study was mostly conducted during school breaks and some during their clinic appointment dates. Scheduling with other participants was based on their availability. Hence, participation in the interviews did not have any interference with the participant’s schedule/activities.

Were there any pilots performed? If so, should be mentioned.

We did not perform pilot studies 

What proportion of those approached agreed to participate, if known? If the number of people invited is unknown, this should be stated, and why it is unknown.

Among the 33 AYA who were approached to participate in the in-depth interviews, 30 participated. Two were not interested whereas one did not show up for the interview. However, all participants mobilized for FGDs participated (Line 174-177).

I am not clear whether the AYAs and the carers were known to each other or not, i.e. were the carers picked because they cared for the specific AYA being interviewed?

They were not necessarily parents of AYAs recruited in the study, but they were known to each other. 

How were carers and healthcare workers approached to be in the study, I couldn’t see this?

A list generated from the adolescent reception module contained contacts of all AYA’s parents/caregivers. The staff used the list to approach parents who accompanied the AYA’s during clinic visit and called those who did not attend clinic. The clinic in charges approached the healthcare workers in each department and informed them about the study. Those interested were referred to the Research Assistants and the FGDs were scheduled. (Line 167-172)

How was the number 30 chosen? I am not clear at what point saturation was found – was it after all 30 had been interviewed or before then? The wording isn’t quite clear. It reads more like 30 was chosen as it was expected that by this point saturation would have been reached, which is not quite the same thing as reaching thematic saturation.

Initially we were approved to conduct up to 40 in-depth interviews and 16 FGDs. However, we reached theoretical saturation after 30 IDIs and 8 FGDs. 

Were all 3 FGDs asking the same questions, or was each group tackling different questions? Also, how many people were in each FGD?

All the three FGDs asked similar questions. 

Results 

What groups were the healthcare workers from? E.g. what proportion was nursing, doctors, etc.

The healthcare workers included were nurses (18.75%), clinical officers (18.75%), Pharmacy technologists (12.5%), Laboratory technicians (12.5%), Community health Assistants (18.75%), and Peer Educators (18.75%). Line 278-280 updated

What proportion of the patients were living with HIV? 

All adolescents who participated in the IDI and FGD were HIV positive and enrolled in care across the two hospitals. However, we did not collect information on the HIV status of the caregivers and healthcare workers.

And what proportion of the patients participated in group discussions? Did all the AYAs have IDIs and FGDs? 

The two groups were mutually exclusive. Those who participated in the IDIs were not invited to participate in FGDs and vice versa.

At least one male in the staff group seems to have (perhaps understandably as presumably he was indicating that he is a father) said that he was previously pregnant/expectant – this raises questions about whether the question was relayed to all participants in the same way? It is probably worth unpicking from your data and commenting to explain how this happened (or changing the way the question is described in the table). 

Thank you for this comment. The question asked was, “How do people react if a young woman becomes pregnant/a young man becomes a father?” The question was relayed to the participants the same way. The said participant meant he was a father.

I would have liked a general paragraph in the results to outline how the FGDs had been – was there discord? How much did people disclose? Did boys or girls, older or younger people dominate discussions in the FGD for AYAs? Were staff in general agreement or were there differences eg. based on rank or location of their work? Was there a lot of difference in the views raised at the 3 groups? Were the caregivers able to give many opinions? What were the main concerns within the 3 groups of people?

We did not observe any noticeable differences between the views of boys and girls, caregivers and healthcare providers-subgroup analysis. We have included this statement on line 281-282.

Why are the IDIs so underrepresented in your quotes? It seems a shame that you did ?30 IDIs and they only get 2 quotes. 

We have included additional quotes from the IDIs in the results section. See line 300 - 315, 321-325, 450-453, and 480-486.

Preferences sections:

Were the locations discussed (schools, hotels, pharmacies etc) prompted from a list/interview or FG prompt or spontaneously mentioned by the participants in the different groups?

The responses were mostly spontaneous based on the question “where do you prefer to receive SRH services and why?”

“If you access those other things, maybe there are not good places you can get from them. But here, they will tell you the good ones and the bad ones (IDI #2 at KCH)” I think this quote needs some extra words in square brackets to help it make sense as at the moment I don’t understand it, even within the context given in the text

We have included additional quotes from the participant and an explanation in brackets to give context. (Line 350-355) 

“I feel that it would be beneficial because I have seen community health volunteers who talk to children, and they open up. I think the issue of age makes them have confidence in you (FGD on 23 August 2019” – this quote needs to say which of the 3 FGD groups this quote came from-specify if adolescent female, male, healthcare provider or carer

The quote came from the FGD with caregivers. We have updated this information (Line 341).

“Others were inclined to have a trained professional go to schools on specific days to provide SRH services and offer psychosocial advice. However, other participants had a contrary opinion arguing that if teachers were to be assigned to provide the services, they would be judgmental and deny AYAs services with the reasoning that they are young and they should not be accessing such services.” - I do not find these two statements contradictory unless one assumes that the trained professional going to the schools is a teacher (and I don’t know why they would necessarily be?). there is also quite a lot of duplication of this paragraph in the following quotation so it could be trimmed.

We have revised the paragraph based on the suggestion to read as below: 

Others were inclined to have a trained professional go to schools on specific days to provide SRH services and offer psychosocial advice. Others argued that if teachers were to be assigned to provide the services, they would be judgmental and deny AYAs services because they were young. 

“elderly” doesn’t really feel like the correct word, I think, where you have used it. Perhaps just “older” would be better?

Thank you for the suggestion. We have replaced the word “elderly” with “older” in all sentences we have used it.

Throughout the section on preferences of counsellors I wasn’t getting a very good idea of which groups of people were thinking what in terms of the preferences. This needs a bit more clarity (even if they were all in agreement, for example).

We have specified the specific group in terms of preferences and stated that the preferences cut across the three groups (AYA, healthcare providers, and caregivers) during IDIs and FGDs. We also included additional quotes to illustrate. 

“receiving services at the facility” – this could be clearer eg. by stating “within a hospital building” (as to me, facility could mean anything including a school for example). I think this terminology is in the abstract as well and would just be clearer if it used more precise language.

We have replaced the word facility with hospital and HIV clinic as appropriate.

Key point: Did the IDIs concur up with the FGDs on your key findings (ie. preferred gender and age of care provider for girls and boys, and preference for SRH services co-situated at a venue like a clinic or hospital)? Did the different FGDs generally concur with this or were some more in agreement with your conclusion and others less so? I think this needs a bit more fleshing out as it is interesting if there are trends (rather than just the odd quote).

The IDIs concurred with the FGDs on key findings. There were no noticeable differences. 

Discussion:

I don’t feel from reading your results that there was a strong preference for hospital or clinic co-situated SRH facilities, among the AYAs; rather that there was a large mix of views and maybe the predominant view was one of reference for co-situated SRH services (the word you use is “many”). I feel the discussion and abstract implies there was a fairly clear preference for healthcare co-situated SRH services. Therefore I suggest that the results section needs to justify the conclusion that there was an overall preference (eg. by stating that a majority of participants said it (please also see above note about showing more clearly if it was in FGD or in IDIs that this conclusion was most evident; AYAs or staff/caregivers)), or the discussion/abstract need to decrease the weight currently placed on preference for SRH being co-situated with healthcare (eg by saying the most commonly voiced preference by AYAs in ___(FGD/IDIs) was x”).

Thank you for this comment. We have revised the discussion section to capture the most common voiced preference by the AYA’s and included additional quotes for illustration.

“AYAs generally preferred receiving SRH services from the health facilities offered and served by trained healthcare providers.” – having read the whole paper, I am not sure I have seen evidence of the “trained healthcare providers” bit of this statement? I think maybe the abstract should just say that AYAs generally preferred SRH services to be co-situated with clinical facilities? Your data seems to support the idea that healthcare providers feel that training is important. 

Thank you for this comment. Based on the findings, we have revised the abstract and manuscript to say that AYAs generally preferred receiving SRH services co-situated with clinical facilities (Line 432-433).

“One of our main findings was the need to design SRH interventions that ensure all services the AYAs require are “under one roof."”. This may be true, but you don’t evidence it much if it is a main finding. For example, there are no quotes from AYAs illustrating this (the quote that mentions parents not knowing what services are being sought does not illustrate the desire for a “one stop shop” with multiple services for the AYA, merely the fact that a health facility masks the SRH activity by also offering non-SRH activity that the person does not intend to use).

We have included quotes from IDIs and FGDs to illustrate that the participants’ preferred receiving all services under one roof at the hospitals.

The quote that mentions parents not knowing what services are being sought illustrates that since the AYA cannot share with the parent/carer regarding the need to access SRH services, going to a hospital with services under one roof allows them to access SRH services in the pretense that they went for treatment of other health conditions. 

The sentences about the youth centers are interesting, but I don’t really see how they link in with the “under one roof” idea very clearly – are they co-sited with HIV clinical teams? This needs greater clarity. 

The youth centers are co-situated within the HIV clinic. These are specifically meant to serve AYA up to the age of 24 years before they transition to adult HIV clinic. Having youth centers with all services under one roof reduces the potential loss of participants as they move from one clinical room to another, and stigma associated with receiving certain services at designated clinical rooms.

I’m not sure what a “stigmatizing outburst” is – needs greater clarity/rephrasing.

We have replaced “stigmatizing outburst” with stigma (Line 589). 

“These findings highlight the importance of having a ‘safe space’ environment for the provision of SRH services within our health facilities and interventions that will improve access to the existing SRH service provided to adolescents living with HIV/AIDs at the adolescent centers.” Who does “our” refer to? And where would the safe spaces be? Within PEPFAR or within the health centers?

“Our” referred to the HIV clinic. We have omitted the word. We meant a conducive environment (safe spaces) within the HIV clinics at the hospitals.

It would be interesting to hear why you think that girls preferred younger, and men older, healthcare professionals – is there any theory you can link with?

From our review of literature most theories we have come across focus on gender preferences, but we have not come across one that specifically points out to why girls would prefer younger and men older. However, we think that girls preferred younger healthcare professionals because they are approachable, the female AYA’s are free to share with them and have recent experience transitioning from adolescent to adulthood. In most cultures from different ethnic communities in Kenya, males are mentored on social roles and community cultures by older men therefore, there is likelihood that the male AYA would consider having an older man for their SRH needs. 

Typos:

AYA, SRH, SSA, IDI, FGD – define when first mentioned, not later in the document

Defined and updated in the manuscript.

“20 million girls aged (15-19 years)” should be “girls (aged 15-19)”

We have updated the manuscript to reflect this change. 

“if they were aged 14-24, living with HIV and on (in) care at the two facilities” [should read IN care]

 We have updated this in the manuscript

I can see that orphan/total orphan is a recognized term, but it is a bit confusing – suggest it might help readability to change to “one/both parents deceased”

Thank you for this suggestion, we have replaced orphan/total orphan with one/both parents deceased.

“This ensured that participants were free to discuss sensitive issues” – suggest you might mean “felt comfortable”?

Yes, meant felt comfortable. The revised sentence reads this, “This ensured that the participants felt comfortable discussing sensitive issues”

Not sure if a typo – what are “health families” p.15 on my document?

We apologize, this was a typo. We meant health facility. We have revised to read hospital. 

“This study explored the perspectives of adolescents, caregivers, and health care providers on improved access to sexual and reproductive health services of AYAs living within western Kenya.” Currently this reads as though the services are already improved and people are being asked about them – I’m not sure this is what you mean though. 

We have replaced the word “improved” with “improving”

“horrible staff attitude” – suggest you mean “disrespectful staff attitudes” or something similar. “Horrible” is not an appropriate word for a journal, unless a quotation.

Thank you for this suggestion. We have replaced “horrible staff attitude” with “disrespectful staff attitudes” 

“In as much as AYAs in our study had a strong preference for receiving their SRH services at health facilities, some preferred other outlets like retail pharmacies” – this doesn’t make sense to me linguistically and I think you mean “Although most of the AYAs in our study had a strong preference for receiving their SRH services at health facilities, some preferred other outlets like retail pharmacies”. Thank you for this suggestion. This was an error. We have revised the sentence to read as below:

“Although most of the AYAs in our study had a strong preference for receiving their SRH services at health facilities, some preferred other outlets like retail pharmacies”

• Reviewer #2: 

Well written and excellent manuscript of an important issue around AYA. The introduction however needs to be revised on various aspects- grammar, references and literature. 

We have revised the introduction section of the manuscript as advised.

The factors outlined as contributing to inadequate access to SRH do not only apply to ALHIV but to all adolescents in general. The distinction between these two groups and the differences needs to be outlines as well as the possible reasons. More literature is needed in the background regarding preferences of healthcare providers and caregiver. More results could be included as well. Also in the discussion comparison with other studies could be made stronger.

Thank you for the suggestions. We have revised the background to include more literature (line 76-85), added more quotes to illustrate our findings in the results section and modifications in the discussion.

---

## [Decision Letter · Decision Letter 1]

19 Oct 2022

PONE-D-21-29137R1Preferences for accessing sexual and reproductive health services among adolescents and young adults living with HIV/AIDs in Western Kenya: A qualitative study.PLOS ONE

Dear Dr. Adhiambo,

Thank you for submitting your manuscript to PLOS ONE. After careful consideration, we feel that it has merit but does not fully meet PLOS ONE’s publication criteria as it currently stands. Therefore, we invite you to submit a revised version of the manuscript that addresses the points raised during the review process. The reviewer(s) note significant improvement in the manuscript but have detailed some additional minor changes that need to be made to further improve the paper. Please respond appropriately to each of these requested changes.

We look forward to receiving your revised manuscript.

Kind regards,

Bettye A. Apenteng

Academic Editor

PLOS ONE

Journal Requirements:

Reviewers' comments:

Reviewer's Responses to Questions

**Comments to the Author**

1. If the authors have adequately addressed your comments raised in a previous round of review and you feel that this manuscript is now acceptable for publication, you may indicate that here to bypass the “Comments to the Author” section, enter your conflict of interest statement in the “Confidential to Editor” section, and submit your "Accept" recommendation.

Reviewer #1: (No Response)

2. Is the manuscript technically sound, and do the data support the conclusions?

Reviewer #1: Yes

3. Has the statistical analysis been performed appropriately and rigorously? 

Reviewer #1: N/A

4. Have the authors made all data underlying the findings in their manuscript fully available?

Reviewer #1: Yes

5. Is the manuscript presented in an intelligible fashion and written in standard English?

Reviewer #1: Yes

6. Review Comments to the Author

Reviewer #1: This is a big improvement, thank you for putting in the time! I especially found the topic guides were helpful, and the additional quotes really brought the topic alive. My only comments are (and apology that they are not in the same order as the issues come up in the text):

1. Table 1 shows a total of 9 female healthcare workers, and also says that 10 healthcare workers had ever been pregnant/expecting - perhaps if the question was "whether you or a partner had ever been pregnant" (is this what you mean by the question "ever been pregnant/expecting"?) was what was interpreted of this question by participants? It's not the biggest point, but your paper will appear more reliable if it doesn't seem to be saying that men were saying they had been pregnant (I'm assuming no transgendered individuals here).

2. As I suggested, please include age characteristics of interviewers in the article text (apologies if I missed it).

3. Please make explicit, as you said to me in response to my comments, that the FGDs and the IDIs concurred with each other around your overall conclusions, and that you did not see any significant differences between the AYAs/caregivers/healthcare workers on where SRH services were best provided. This is an important finding.

4. line 250 - the quote following this sentence about AYAs' expressed preferences seems in fact to have been from an adult, so it doesn't illustrate the expressed preference, only another person's assertion as to their preferences (minor point).

5. Darting back to the methods section, i think that what you told me about how you conducted interviews in school break times to accommodate the AYAs' other needs and to aid their ability to participate is something I would suggest including in the text. This demonstrates good practice for other research groups (I couldn't see it - apologies if it is there).

6. I would like to see your hypotheses for why adolescent boys like getting SRH care from older men and adolescent girls from younger women explored in the document - you have fed back some ideas to me, but not put them as hypotheses in the discussion as far as I can see (apologies if I missed it). Cite evidence where you can, but ultimately this is a potentially interesting avenue for future research, which you should point out. Your standpoint within a similar cultural millieu of the AYAs means that your interpretation is far better than, say, mine would be, and it is interesting and useful!

7. Darting to the interview and FGD guides - were all questions used for all interviews and all FDGs, or did you only say focus on one or two areas for each FDG, say? it seems like a lot of quesitons to get through in quite a short time. Please outline whether all question areas were covered in all interviews and FGDs, or whether the FGDs and interviews tended to pick up on just a few areas. To my mind it doesn't matter which is true, but i think it needs to be stated.

Thank you again, these are my only thoughts now and they are quite minor, great job!

7. PLOS authors have the option to publish the peer review history of their article (what does this mean?). If published, this will include your full peer review and any attached files.

Reviewer #1: No

---

## [Author Response · Author response to Decision Letter 1]

25 Oct 2022

1. Table 1 shows a total of 9 female healthcare workers and also says that ten healthcare workers had ever been pregnant/expecting - perhaps if the question was "whether you or a partner had ever been pregnant" (is this what you mean by the question "ever been pregnant/expecting"?) was what was interpreted of this question by participants? It's not the biggest point, but your paper will appear more reliable if it doesn't seem to be saying that men were saying they had been pregnant (I'm assuming no transgendered individuals here).

The question was asked across both genders. Whether females had ever been pregnant/expectant or if they were male if they had ever made someone pregnant. However, we have amended the table by omitting the proportion of caregivers and health workers who were pregnant/expectant since it doesn't add much value (Table 1 revised).

2. As I suggested, please include age characteristics of interviewers in the article text (apologies if I missed it).

Apologies for the oversight. We responded in the rebuttal letter but failed to include it in the manuscript. We have included it on line 12 and line 145.

3. Please make explicit, as you said to me in response to my comments, that the FGDs and the IDIs concurred with each other around your overall conclusions and that you did not see any significant differences between the AYAs/caregivers/healthcare workers on where SRH services were best provided. This is an important finding.

Thank you for this comment. We have included the statement in our overall conclusion. Line 439-449.

4. line 250 - the quote following this sentence about AYAs' expressed preferences seems to have been from an adult, so it doesn't illustrate the expressed preference, only another person's assertion about their preferences (minor point).

We have replaced the quote with two quotes from AYA IDIs (Lines 261-267)

5. Darting back to the methods section, I think that what you told me about how you conducted interviews during school break times to accommodate the AYAs' other needs and to aid their ability to participate in something I would suggest including in the text. This demonstrates good practice for other research groups (I couldn't see it - apologies if it is there).

Thank you for this comment. We have included this statement on lines 124-127

6. I would like to see your hypotheses for why adolescent boys like getting SRH care from older men and adolescent girls from younger women explored in the document - you have fed back some ideas to me but not put them as hypotheses in the discussion as far as I can see (apologies if I missed it). Cite evidence where you can, but ultimately this is a potentially interesting avenue for future research, which you should point out. Your standpoint within a similar cultural milieu of the AYAs means that your interpretation is far better than, say, mine would be, and it is interesting and useful!

We have included the hypothesis in lines 419-424

7. Darting to the interview and FGD guides - were all questions used for all interviews and all FDGs, or did you only say focus on one or two areas for each FDG, say? it seems like a lot of questions to get through in quite a short time. Please outline whether all question areas were covered in all interviews and FGDs, or whether the FGDs and interviews tended to pick up on just a few areas. To my mind, it doesn't matter which is true, but I think it needs to be stated.

We had guides for each of the two data collection methods, i.e. FGD and IDI. Updated on Lines 152-153

Thank you again, these are my only thoughts now, and they are quite minor, great job!

Thank you so much for the comments; they were very insightful.

---

## [Editor Report · Decision Letter 2]

28 Oct 2022

Preferences for accessing sexual and reproductive health services among adolescents and young adults living with HIV/AIDs in Western Kenya: A qualitative study.

PONE-D-21-29137R2

Dear Dr. Adhiambo,

We’re pleased to inform you that your manuscript has been judged scientifically suitable for publication and will be formally accepted for publication once it meets all outstanding technical requirements.

Kind regards,

Bettye A. Apenteng

Academic Editor

PLOS ONE
---

## [Editor Report · Acceptance letter]

7 Nov 2022

PONE-D-21-29137R2 

Preferences for accessing sexual and reproductive health services among adolescents and young adults living with HIV/AIDs in Western Kenya: *A qualitative study.*

Dear Dr. Adhiambo:

I'm pleased to inform you that your manuscript has been deemed suitable for publication in PLOS ONE. Congratulations! Your manuscript is now with our production department. 

Kind regards, 

on behalf of

Dr. Bettye A. Apenteng 

Academic Editor

PLOS ONE